# IL-4 and helminth infection downregulate MINCLE-dependent macrophage response to mycobacteria and Th17 adjuvanticity

Judith Schick[1], Meltem Altunay[1†], Matthew Lacorcia[2,3†], Nathalie Marschner[1], Stefanie Westermann[4], Julia Schluckebier[2,3], Christoph Schubart[4], Barbara Bodendorfer[1], Dennis Christensen[5], Christian Alexander[6], Stefan Wirtz[7], David Voehringer[4], Clarissa Prazeres da Costa[2,3], Roland Lang[1*]

[1]Institut für Klinische Mikrobiologie, Immunologie und Hygiene, Universitätsklinikum Erlangen, Friedrich-Alexander Universität Erlangen-Nürnberg, Erlangen, Germany; [2]Institut für Medizinische Mikrobiologie, Immunologie und Hygiene, Center for Global Health, Technische Universität München, Munich, Germany; [3]Center for Global Health, Technical University Munich, Munich, Germany; [4]Infektionsbiologische Abteilung, Universitätsklinikum Erlangen, Friedrich-Alexander Universität Erlangen-Nürnberg, Erlangen, Germany; [5]Adjuvant Research, Department of Infectious Disease Immunology, Statens Serum Institut, Copenhagen, Denmark; [6]Cellular Microbiology, Forschungszentrum Borstel, Leibniz Lung Center Borstel, Borstel, Germany; [7]Medizinische Klinik 1, Universitätsklinikum Erlangen, Friedrich-Alexander Universität Erlangen-Nürnberg, Erlangen, Germany

**\*For correspondence:**
roland.lang@uk-erlangen.de

†These authors contributed equally to this work

**Competing interest:** The authors declare that no competing interests exist.

**Abstract** The myeloid C-type lectin receptor (CLR) MINCLE senses the mycobacterial cell wall component trehalose-6,6'-dimycolate (TDM). Recently, we found that IL-4 downregulates MINCLE expression in macrophages. IL-4 is a hallmark cytokine in helminth infections, which appear to increase the risk for mycobacterial infection and active tuberculosis. Here, we investigated functional consequences of IL-4 and helminth infection on MINCLE-driven macrophage activation and Th1/Th17 adjuvanticity. IL-4 inhibited MINCLE and cytokine induction after macrophage infection with *Mycobacterium bovis* bacille Calmette-Guerin (BCG). Infection of mice with BCG upregulated MINCLE on myeloid cells, which was inhibited by IL-4 plasmid injection and by infection with the nematode *Nippostrongylus brasiliensis* in monocytes. To determine the impact of helminth infection on MINCLE-dependent immune responses, we vaccinated mice with a recombinant protein together with the MINCLE ligand trehalose-6,6-dibehenate (TDB) as adjuvant. Concurrent infection with *N. brasiliensis* or with *Schistosoma mansoni* promoted T cell-derived IL-4 production and suppressed Th1/Th17 differentiation in the spleen. In contrast, helminth infection did not reduce Th1/Th17 induction by TDB in draining peripheral lymph nodes, where IL-4 levels were unaltered. Upon use of the TLR4-dependent adjuvant G3D6A, *N. brasiliensis* infection impaired selectively the induction of splenic antigen-specific Th1 but not of Th17 cells. Inhibition of MINCLE-dependent Th1/Th17 responses in mice infected with *N. brasiliensis* was dependent on IL-4/IL-13. Thus, helminth infection attenuated the Th17 response to MINCLE-dependent immunization in an organ- and adjuvant-specific manner via the Th2 cytokines IL-4/IL-13. Taken together, our results demonstrate downregulation of MINCLE expression on monocytes and macrophages by IL-4 as a possible mechanism of thwarted Th17 vaccination responses by underlying helminth infection.

## Editor's evaluation

The effect of helminth infection on vaccination against tuberculosis infection and disease is an important area of study. In this manuscript, the authors build off of a large body of prior data showing that mycobacterial antigens upregulate MINCLE whilst the cytokine IL-4 downregulates MINCLE, and as IL-4 is upregulated during Helminth infections, this can antagonize Th1/Th17 responses. By using two different models of helminth infection, the authors demonstrate compelling organ-specific impairment of Th17 responses in a vaccination setting with a MINCLE-dependent adjuvant. The work is topical, may have important translational implications for patients with tuberculosis and helminth co-infections and/or vaccination regimens for patients with helminth infections, and will be of interest to individuals studying the convergence of different infectious diseases.

## Introduction

Tuberculosis kills more than 1 million humans each year and is easily spread by aerosol infection. Importantly, after exposure to *Mycobacterium tuberculosis* (MTB) by inhalation, only a minority of contacts develops active pulmonary disease with mycobacterial replication and destruction of lung tissue. In the majority of exposed people, inhaled MTB is either killed by the innate immune system of the lung or triggers the development of granuloma structures around infected cells (*Pai et al., 2016*). The development of specific CD4+ T cell responses then induces control of mycobacterial growth without achieving eradication of MTB, a status referred to as latency (*Pai et al., 2016*). It is estimated that more than a quarter of the world's population is latently infected with MTB. Their life-long risk to develop reactivation tuberculosis is around 5% and is influenced by a variety of genetic and environmental factors.

Parasitic helminth infections with intestinal nematodes, filaria, or trematodes affect an estimated number of >2 billion humans (*Babu and Nutman, 2016*). Similar to tuberculosis, helminth infections are often chronic and not acutely life-threatening. Epidemiologically, tuberculosis and helminth infections co-occur in many tropical and subtropical countries (*Salgame et al., 2013*). Immune responses to helminths in general are characterized by a strong Th2 component with production of IL-4, IL-5, IL-13, IL-10, and IgE (*Babu and Nutman, 2016*), whereas protective immunity to MTB requires robust Th1 immunity and is possibly enhanced by Th17 cells (*Gopal et al., 2012*; *Monin et al., 2015*). These diametrically opposed immunological biases and effector mechanisms raise the question how MTB and parasitic helminths mutually influence each other in co-infected individuals (*Babu and Nutman, 2016*). Interestingly, helminth infection of household contacts of patients with active, smear-positive tuberculosis increased the rate of tuberculin skin test conversion indicating a higher risk to become infected (*Verhagen et al., 2012*). Furthermore, active tuberculosis patients are more often co-infected with helminths than healthy counterparts (*Elias et al., 2006*; *Tristão-Sá et al., 2002*). Furthermore, helminth co-infection is associated with more advanced disease in tuberculosis patients and with reduced production of IFNγ but increased IL-10 (*Resende Co et al., 2007*). These findings indicate that the type 2 immune bias in helminth-infected individuals can impair the development of protective adaptive immunity to MTB. In contrast, other studies showed that underlying helminth infection can promote a Th2 and regulatory T cell response to MTB with a reduced risk to develop open, smear-positive pulmonary tuberculosis (*Abate et al., 2015*). In mice, infection with *Nippostrongylus brasiliensis* enhances bacterial load in the lung after MTB infection, an effect attributed to the action of alternatively activated macrophages (*Potian et al., 2011*).

Underlying helminth infection may also influence the response to vaccination with *Mycobacterium bovis* bacille Calmette-Guerin (BCG), which shows large geographic variation with reduced efficacy in countries with a high prevalence of worm infections (*Roy et al., 2014*; *Colditz et al., 1994*). Experimental infection of mice with *Schistosoma (S.) mansoni* or *Heligmosomoides (H.) polygyrus* impaired Th1 cell generation after vaccination with BCG (*Elias et al., 2005*; *Obieglo et al., 2016*), although another study found that chronic enteric infection with *H. polygyrus* did not interfere with primary or memory T cell responses to BCG (*Rafi et al., 2015*). Thus, these results from experimental mouse models show that not all helminth infections have a similarly strong impact on protective immunity to MTB. Anthelminthic treatment prior to BCG administration led to increased IFNγ and IL-12 production (*Elias et al., 2001*). It is noteworthy that maternal helminth infection during pregnancy negatively affected the frequency of IFNγ-producing T cells in cord blood of

neonates (*Gebreegziabiher et al., 2014*) and the development of Th1 immunity in BCG-vaccinated offspring (*Malhotra et al., 1999*). However, attempts to enhance the immunogenicity of BCG vaccination by anthelminthic treatment of mothers during pregnancy were not successful (*Webb et al., 2011*).

Sensing of mycobacteria by the innate immune system is essential for a robust defense reaction, including the secretion of chemokines to attract leukocytes to the site of infection and the production of cytokines directing antigen-specific T cell responses toward Th1 and Th17 differentiation. Mycobacteria contain a plethora of PAMPs, for example, the TLR ligands 19 kDa lipoprotein, lipoarabinomannan (LAM), and mycobacterial DNA rich in CpG motifs (*Mortaz et al., 2015*). The hydrophobic cell wall of mycobacteria is especially rich in the glycolipid trehalose-6,6'-dimycolate (TDM, *aka* the cord factor), which is sufficient to trigger granulomatous reactions in vivo (*Ishikawa et al., 2009*), induces inflammatory cytokine production by macrophages (*Geisel et al., 2005*), and confers the Th1/Th17 adjuvant activity known from Freund's complete adjuvant (*Shenderov et al., 2013*). TDM binds to the C-type lectin receptor (CLR) MINCLE (encoded by the *Clec4e* gene) (*Ishikawa et al., 2009*; *Schoenen et al., 2010*), which is inducibly expressed in macrophages (*Matsumoto et al., 1999*) and associates with the adapter protein Fc receptor gamma chain (FcRγ, encoded by *Fcer1g*) (*Yamasaki et al., 2008*). Signalling by MINCLE requires binding of the kinase SYK to the phosphorylated ITAM of the FcRγ, and proceeds via the CARD9-BCL10-MALT1 complex to activation of NFκB and expression of inflammatory target genes (*Werninghaus et al., 2009*; *Ostrop and Lang, 2017*). In addition to TDM, several additional glycolipids from the mycobacterial cell wall were recently identified as ligands for CLR, most of them like MINCLE members of the FcRγ-coupled DECTIN-2 family (*Ishikawa et al., 2017*). MCL, closely related to MINCLE and able to heterodimerize with it, also can bind TDM and in addition glycero-monomycolate (*Furukawa et al., 2013*; *Miyake et al., 2013*); DECTIN-2 binds mannose-capped LAM (*Yonekawa et al., 2014*) and DCAR is a receptor for mycobacterial phosphatidyl-inositol mannosides (*Toyonaga et al., 2016*). The prototypic myeloid signaling CLR, DECTIN-1 directly recruits SYK through its non-classical ITAM (*Rogers et al., 2005*) it, too, binds and is activated by an yet unidentified mycobacterial ligand (*Rothfuchs et al., 2007*). Since the signals emanating from the interaction of multiple CLR with MTB are transmitted through SYK-CARD9, it is not surprising that CARD9-deficient mice are highly susceptible to infection with MTB and succumb with high bacterial burden in the organs (*Dorhoi et al., 2010*). Knockout mice for individual CLR showed less severe phenotypes but were, dependent on the infection model employed, moderately more susceptible to infection (*Wilson et al., 2015*; *Behler et al., 2015*; *Behler et al., 2012*; *Lee et al., 2012*).

Already in the first report on MINCLE, the inducible expression in response to TLR stimulation with LPS and by the cytokine TNF was reported (*Matsumoto et al., 1999*). Indeed, TNF is essential for the upregulation of MINCLE in macrophages stimulated with the cord factor and for the Th17 promoting effect of the adjuvant CAF01 that contains the MINCLE ligand TDB (*Schick et al., 2020*). In contrast, the Th2 cytokines IL-4 and IL-13 downregulate expression of MINCLE, MCL, and DECTIN-2, in human monocytes/macrophages and DC, as well as in mouse bone marrow-derived macrophages (*Hupfer et al., 2016*; *Ostrop et al., 2015*). This inhibitory effect of the Th2 cytokines IL-4 and IL-13 depends on the transcription factor STAT6 and is associated with a reduced production of G-CSF and TNF by macrophages stimulated with the MINCLE ligand TDB, but not after stimulation with LPS (*Hupfer et al., 2016*). Thus, expression of the DECTIN-2 family CLR MINCLE, MCL, and DECTIN-2 is subject to a substantial regulation by the cytokines TNF and IL-4, which are key factors in type 1 and type 2 immune responses, respectively. Since type 2 immunity is a hallmark of many helminth infections, the question arises whether the expression and function of DECTIN-2 family CLR may be compromised during helminth infection in vivo (*Lang and Schick, 2017*).

Here, we have investigated the functional impact of IL-4 and helminth infection on myeloid responses to mycobacteria or the cord factor-based adjuvant TDB in vitro and in vivo. Our data show that IL-4 impairs MINCLE expression on monocytes in vivo, but does not interfere with phagocytosis of BCG. Underlying helminth infection attenuated immune responses to recombinant protein together with different adjuvants in the spleen but not in draining lymph nodes. The specific impairment of Th17 responses to MINCLE-dependent adjuvant indicates the contribution of downregulation of DECTIN-2 family CLR to vaccine antagonism induced by helminth infection.

## Results

### IL-4 impairs upregulation of MINCLE and other DECTIN-2 family CLR in macrophages stimulated with BCG

In previous work, we observed that IL-4 and IL-13 downregulate expression of MINCLE, MCL, and DECTIN-2 when macrophages were stimulated with the MINCLE ligand TDB, but not after triggering TLR4 with LPS (*Hupfer et al., 2016*). Since mycobacteria contain multiple ligands for CLR and TLR, we were interested whether IL-4 has any impact on changes in DECTIN-2 family CLR expression in macrophages stimulated with *M. bovis* BCG (*Figure 1*). Stimulation of bone marrow-derived macrophages (BMM) with BCG caused the strongest induction of MINCLE (up to 400-fold after 48 hr) and was reduced significantly by IL-4 (*Figure 1A*). A similar pattern of inducibility and regulation by IL-4 was observed for DECTIN-2 and MCL, albeit the increases at the mRNA level were more moderate (e.g. 50-fold for DECTIN-2 and 8-fold for MCL after stimulation with BCG) (*Figure 1B and C*). IL-4 did not inhibit, but actually increased expression of DECTIN-1 (*Figure 1D*), confirming a selective impact on DECTIN-2 family CLR. The inhibitory effect of IL-4 on BCG-induced expression of MINCLE and DECTIN-2 was also evident at the protein level, when analysed by flow cytometry (*Figure 1E and F*). Regulation of MCL could not be validated at the protein level because no antibody for flow cytometry was available. DECTIN-1 surface protein levels were increased by IL-4 (*Figure 1G*) consistent with the quantitative real time PCR (qRT-PCR) data and previously published results (*Willment et al., 2003*).

### IL-4 does not affect phagocytosis of BCG, but inhibits cytokine production

Several CLR, including DECTIN-2 and MCL, are phagocytic receptors (*Ifrim et al., 2014*; *Graham et al., 2012*). Therefore, we next measured phagocytosis of fluorescent BCG (expressing DsRed) by flow cytometry in the presence or absence of IL-4 (*Figure 2A*). BMM infected with BCG-DsRed in different MOI dose-dependently ingested the mycobacteria, IL-4 had no impact on phagocytosis after 6 and 24 hr (*Figure 2A*). We next determined whether phagocytosis of BCG requires MINCLE or FcRγ, the common adapter protein of the CLR MINCLE, DECTIN-2, and MCL. Uptake of fluorescent BCG was comparable to WT cells in *Clec4e*$^{-/-}$ and *Fcer1g*$^{-/-}$ BMM (*Figure 2B*). In contrast, the high levels of the cytokines G-CSF and TNF in supernatants of BMM infected with BCG were significantly reduced by co-treatment with IL-4 (*Figure 2B*), extending our previous results for TDB (*Hupfer et al., 2016*) to the response induced by whole mycobacteria. Stimulation of BMM deficient in MINCLE or in the FcRγ chain with BCG led to a reduced production of TNF and G-CSF, confirming the contribution of this receptor to macrophage activation by mycobacteria (*Ishikawa et al., 2009*; *Schoenen et al., 2010*; *Figure 2B*). Of interest, the inhibitory effect of IL-4 was still visible in MINCLE-deficient BMM for G-CSF but not for TNF; in addition, it was much less pronounced than in WT BMM (*Figure 2B*), indicating that downregulation of MINCLE is a prominent mechanism of IL-4-induced impairment of macrophage cytokine production to mycobacteria. Reduced cytokine production in *Clec4e*$^{-/-}$ and *Fcer1g*$^{-/-}$ BMM was not due to a defect in the uptake of BCG (*Figure 2C*).

### Overexpression of IL-4 inhibits MINCLE expression after intraperitoneal infection with BCG

Given the impact of IL-4 on BCG-induced MINCLE expression in vitro, we next used intraperitoneal infection of mice with BCG for analysis of MINCLE regulation by IL-4 in vivo. Hydrodynamic injection of minicircle DNA encoding IL-4 was used to overexpress IL-4 from hepatocytes in vivo (*Suda and Liu, 2007*). Serum levels of IL-4 after injection of 0.25 and 0.5 µg IL-4-encoding minicircle DNA ranged between 1 and 2 ng/ml, whereas IL-4 was not detectable in control mice (*Figure 3A*). Peritoneal lavage cells were obtained 24 hr after i.p. infection with 4×10$^7$ CFU BCG and MINCLE expression on Ly6C$^{hi}$ monocytes and Ly6G$^+$ neutrophils was analysed by flow cytometry (*Figure 3B*, gating strategy). Compared to the PBS control mice, MINCLE cell surface expression increased after BCG infection in monocytes and in neutrophils, which was abrogated by IL-4 overexpression specifically in monocytes but not neutrophils (*Figure 3C*).

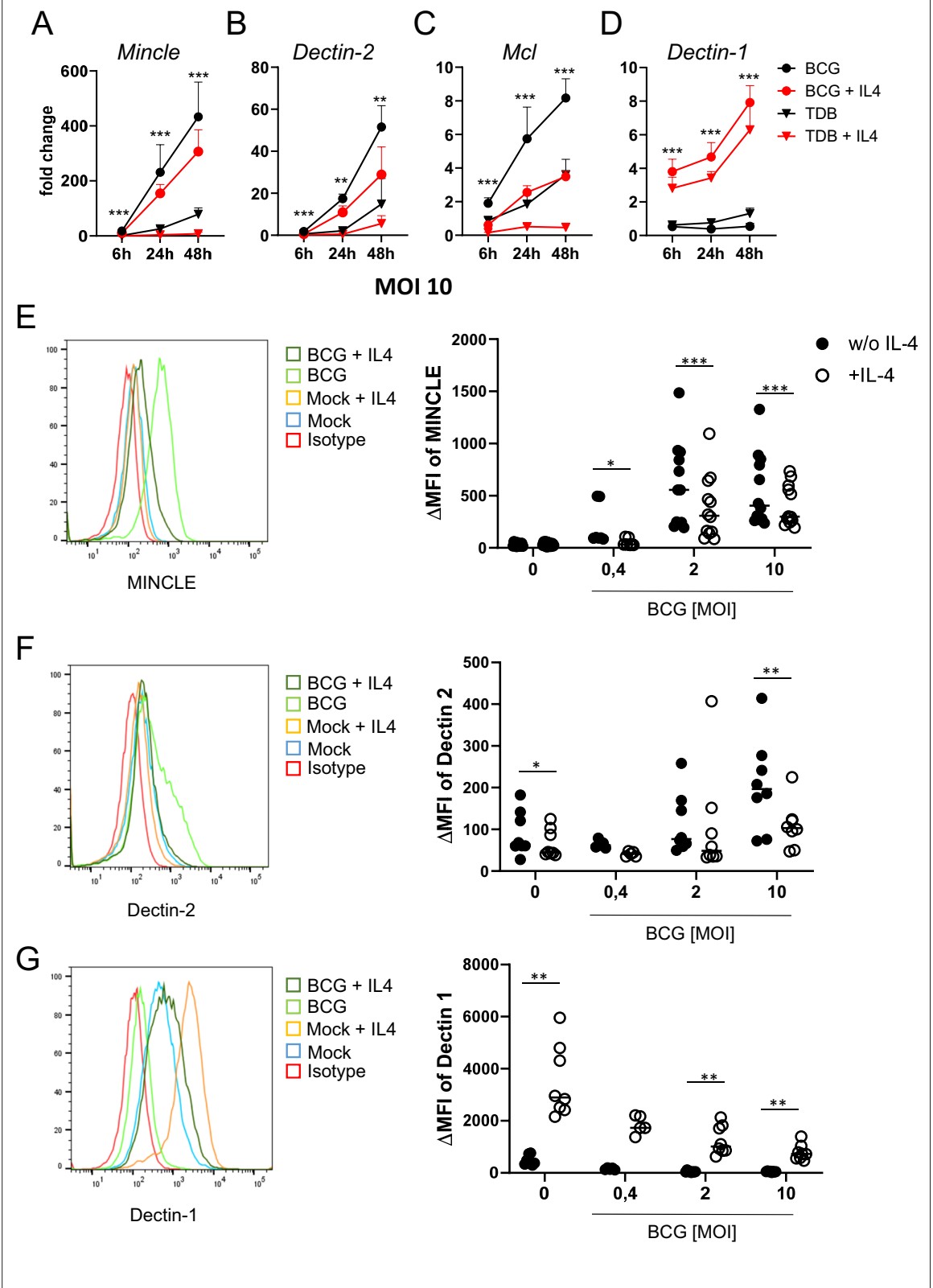

**Figure 1.** IL-4 impairs upregulation of MINCLE and other DECTIN-2 family C-type lectin receptor (CLR) in macrophages stimulated with bacille Calmette-Guerin (BCG). (**A–D**) C57BL/6 bone marrow-derived macrophages (BMM) were stimulated as indicated in presence or absence of IL-4 for 6, 24, or 48 hr. BCG was used at MOI 10. MINCLE (**A**), DECTIN-2 (**B**), MCL (**C**), and DECTIN-1 (**D**) mRNA expression was determined by quantitative real-time PCR (qRT-PCR) shown as fold change calibrated to unstimulated control. Data are depicted as mean + SD from two independent experiments

*Figure 1 continued*

performed in biological duplicates. (**E–G**) BMM were stimulated with BCG at the indicated MOI in the presence or absence of IL-4 for 24 hr, followed by staining for MINCLE (**E**), DECTIN-2 (**F**), or DECTIN-1 (**G**) expression. Representative stainings are shown as histogram overlay (left panel), for quantification (right panel) the median fluorescence intensity of the isotype control staining was subtracted from the respective CLR signal to obtain the ΔMFI. Each point represents one mouse, pooled from at least two independent experiments. *p<0.05, **p<0.01, ***p<0.001 in Wilcoxon signed rank test.

The online version of this article includes the following source data for figure 1:

**Source data 1.** Source data *Figure 1* (qRT-PCR and flow cytometry analysis of CLR in macrophages).

## Co-infection with *N. brasiliensis* impairs MINCLE upregulation on peritoneal monocytes, but does not reduce phagocytosis, upon BCG infection

Infection with the hookworm *N. brasiliensis* induces a strong Th2 response characterized by high levels of IL-4. We therefore asked whether pre-existing infection with *N. brasiliensis* interferes with MINCLE expression on myeloid cells during mycobacterial infection. The peritoneal cavity of *N. brasiliensis*-infected mice contained a much higher proportion of SiglecF$^+$ eosinophils (nearly 40% compared to 10% in controls), whereas the fraction of Ly6C$^{hi}$ monocytes was reduced (from 30% to 20%), and the low percentage of resident monocytes/macrophages was not altered by helminth infection (*Figure 3D*). Eosinophils were negative for MINCLE surface staining. MINCLE staining on inflammatory monocytes from BCG-infected mice was significantly reduced when co-infected with *N. brasiliensis* compared to control mice infected only with BCG (*Figure 3E*). In contrast, the cell surface expression of MINCLE on neutrophils was not altered by underlying *N. brasiliensis* infection (*Figure 3E*). The use of fluorescent BCG-DsRed enabled us to determine the cell types and percentages of peritoneal cells that had ingested mycobacteria 24 hr after injection (*Figure 3F*). While on average 12% of Ly6C$^{hi}$ monocytes contained BCG-DsRed, no specific signal was measurable for the peritoneal neutrophils. Co-infection with *N. brasiliensis* did not change the phagocytosis of BCG by inflammatory monocytes in the peritoneal cavity (*Figure 3F*).

## Co-infection with *N. brasiliensis* or with *S. mansoni* suppresses Th1/Th17 induction by a MINCLE-dependent adjuvant in the spleen

Downregulation of MINCLE expression on monocytes of mice with *N. brasiliensis* infection led us to ask whether vaccination responses to protein antigen induced by a MINCLE-dependent adjuvant would be inhibited by helminth infections. We investigated this question in chronic (non-transient) *S. mansoni* and acute (transient) *N. brasiliensis* infection models. Infected and control mice were immunized with the recombinant fusion protein H1, comprising the MTB antigens Ag85B and ESAT-6, adsorbed to the adjuvant CAF01 (a combination of the MINCLE ligand TDB incorporated into cationic liposomes) (*Agger et al., 2008*). We and others previously showed that subcutaneous injection of the CAF01 adjuvant with H1 induces robust antigen-specific production of IFNγ and IL-17 by CD4+ T cells (*Desel et al., 2013*; *Lindenstrøm et al., 2009*; *Lindenstrøm et al., 2012*). Similar results were obtained for the combination of CAF01 with the chlamydial antigen CTH522 (*Nguyen et al., 2020*).

*S. mansoni* causes a chronic infection in mice, with fertile, adult worms residing in the portal vein system. Trapping of eggs in tissues (especially the liver) is a strong stimulus for Th2-type immunity, which dominates in the later phase of infection after 8–9 weeks. We therefore immunized mice infected with *S. mansoni* in this chronic phase subcutaneously with H1/CAF01 in the flank of the mice (*Figure 4A*). We first analysed whether schistosomal infection indeed enhanced IL-4 production by T cells (*Figure 4B*). While only very low levels of IL-4 were detectable in the supernatants of draining lymph node cells, splenocytes of *S. mansoni*-infected mice produced significant amounts of IL-4 when T cells were polyclonally stimulated with anti-CD3 in vitro. Thus, *S. mansoni* infection generated a Th2 milieu in the spleen, but not in the inguinal lymph node. Induction of antigen-specific Th1 and Th17 cells, detected by robust secretion of IFNγ and IL-17, respectively, was observed after immunization in both lymph nodes and spleen (*Figure 4C and D*). Infection with *S. mansoni* had no impact on H1-specific or anti-CD3-induced production of IFNγ, IL-17, and of IL-10 from draining inguinal lymph nodes (*Figure 4C*). In contrast, the antigen-specific splenocyte responses to H1 were strongly reduced in the case of IFNγ, IL-17, and also for IL-10 (*Figure 4D*). *S. mansoni* infection also suppressed IFNγ, but not

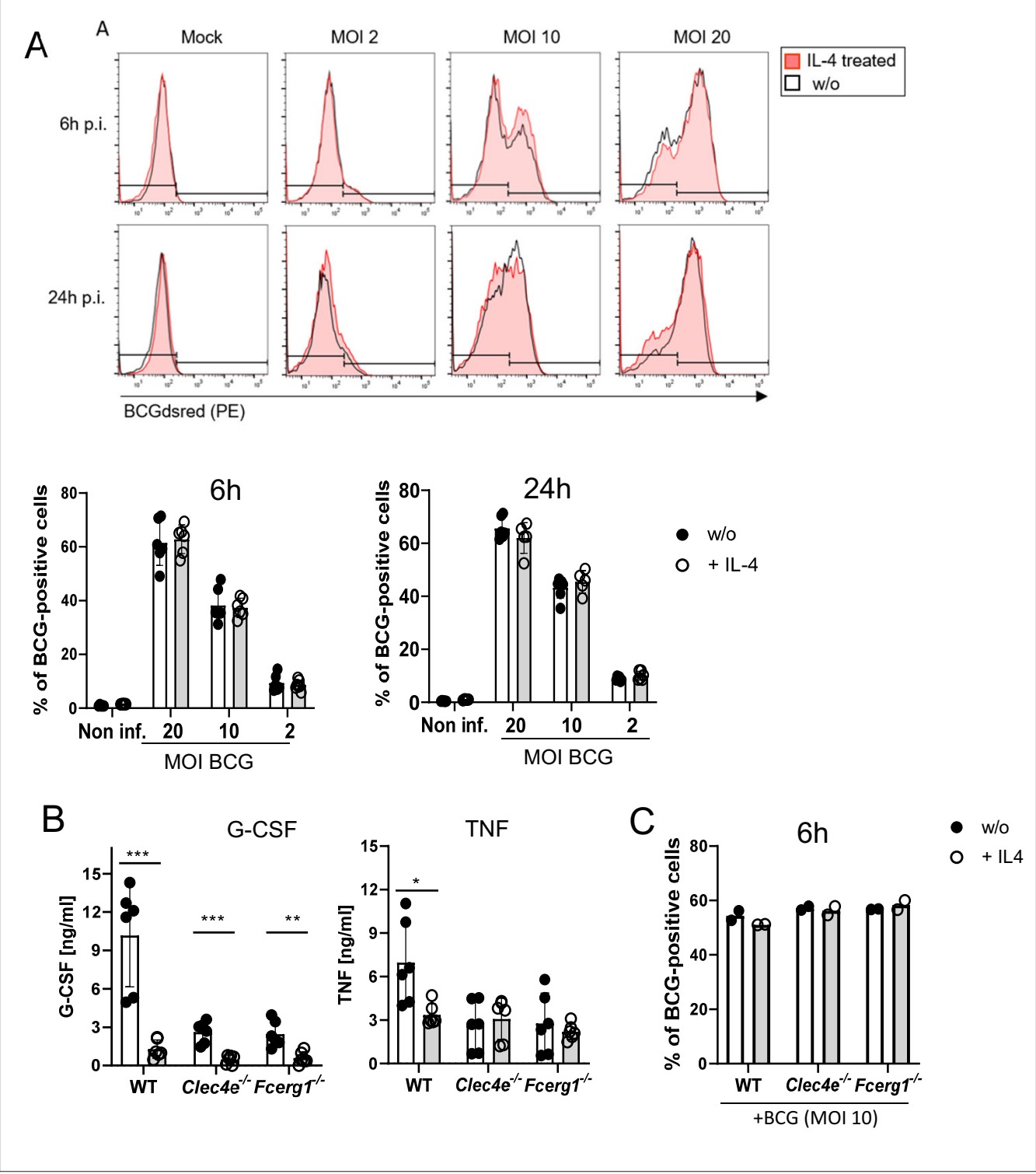

**Figure 2.** IL-4 does not affect phagocytosis of bacille Calmette-Guerin (BCG) but inhibits cytokine production. (**A**) C57BL/6 bone marrow-derived macrophages (BMMs) were infected with different MOI of fluorescent BCG-Dsred co-treated with IL-4 or not as indicated. Phagocytic uptake was measured via flow cytometry. (**A**) Representative histograms show phagocytosis of BCG by detection of fluorescent DsRed signal in BMMs. Quantitative analysis of phagocytosis based on percentage of BCG-positive cells at 6 and 24 hr post infection. Data is depicted from two to three independent experiments performed in biological duplicates. (**B**) C57BL/6 (*WT*), MINCLE knockout (*Clec4e⁻/⁻*), and FcRγ (*Fcer1g⁻/⁻*) BMM were stimulated with BCG for 24 hr. Production of G-CSF and TNF was measured from cell culture supernatants via ELISA. No significant cytokine production was detected from non-stimulated BMM. Data is depicted from three independent experiments performed in biological duplicates. (**C**) Phagocytosis of fluorescent BCG is

*Figure 2 continued on next page*

Figure 2 continued

unaltered in BMM deficient in MINCLE (*Clec4e$^{-/-}$*) or FcRγ (*Fcer1g$^{-/-}$*). Mean percentages of one representative experiment out of two (*Fcer1g$^{-/-}$*) and three (*Clec4e$^{-/-}$*) performed. *p<0.05, **p<0.01, ***p<0.001.

The online version of this article includes the following source data for figure 2:

**Source data 1.** Source data *Figure 2* (phagocytosis and cytokine production).

IL-17 production triggered by polyclonal anti-CD3 stimulation, whereas secretion of IL-10 was much higher from splenocytes of *S. mansoni*-infected mice (*Figure 4D*). Together, these data show that *S. mansoni* infection establishes a Th2 environment in the spleen but not in the inguinal lymph nodes, which corresponds to suppression of Th1 and Th17 responses after CAF01-adjuvanted immunization in splenocytes but has no impact on draining lymph node cells.

To determine whether *S. mansoni*-induced inhibition of antigen-specific immune responses by splenocytes, but not draining lymph node cells, was specific for the MINCLE-dependent adjuvant CAF01, we immunized mice with H1 together with the TLR9 ligand CpG ODN 1826 as adjuvant. Using the same protocol as before for H1/CAF01, we observed, as demonstrated previously (*Werninghaus et al., 2009*), strong induction of IFNγ-producing T cells but a lack of IL-17 production upon re-stimulation of draining lymph node cells or splenocytes (*Figure 4E and F*). Thus, it was not possible to compare inhibition of Th17 responses by helminth infection between adjuvants. However, *S. mansoni* did not generally downregulate IL-17 production from T cells, as the levels after anti-CD3 stimulation were unaffected (*Figure 4E and F*). Similar to the impact on immunization with CAF01, *S. mansoni* infection strongly reduced antigen-specific and non-specific IFNγ secretion by splenocytes but not by draining lymph node cells (*Figure 4E and F*). Thus, *S. mansoni* infection caused a general suppression of the splenic Th1 response to immunization, regardless whether the MINCLE-dependent adjuvant CAF01 was used or the TLR9-dependent CpG ODN 1826.

To address the effect of transient helminth infection, mice were injected subcutaneously with L3 larvae of *N. brasiliensis* and were immunized 5 days later with H1/CAF01 in the footpads of both hindlegs (*Figure 5A*). The local tissue swelling to subcutaneous vaccination in the footpad was not changed in mice with underlying *N. brasiliensis* infection (*Figure 5B*). The total number of cells in the draining inguinal lymph nodes 7 days after immunization was strongly increased compared to non-vaccinated mice, but significantly reduced in mice infected with *N. brasiliensis*; in contrast, the number of splenocytes was unaltered by helminth infection (*Figure 5C*). As was already observed after infection with *S. mansoni*, establishment of IL-4-producing Th2 T cells was only detected in spleens of *N. brasiliensis*-infected mice but not in their popliteal and inguinal lymph nodes (*Figure 5D*). Upon re-stimulation of lymph node cells with H1 in vitro, the antigen-specific production of IFNγ, IL-17, or IL-10 was not affected by *N. brasiliensis* co-infection (*Figure 5E*). In striking contrast, splenocytes from co-infected mice generated significantly less IFNγ and IL-17, but not IL-10, when re-stimulated with H1 antigen, whereas for polyclonal stimulation with anti-CD3 antibody only a reduction for IFNγ was observed (*Figure 5F*). When mice were immunized with H1 together with the TLR4-dependent adjuvant G3D6A, IFNγ production by splenocytes was diminished in mice infected with *N. brasiliensis*, whereas antigen-specific secretion of IL-17, IL-10, and IL-4 were not changed (*Figure 5G*). However, polyclonal stimulation of T cells from the spleens of *N. brasiliensis*-infected mice caused secretion of significant IL-4 (*Figure 5G*), as after immunization with the MINCLE-dependent adjuvant CAF01 (*Figure 5D*).

Together, these results indicate that underlying *N. brasiliensis* infection attenuated cell expansion in the draining lymph node, yet the differentiation toward Th1/Th17 at this site was not affected. In contrast, hookworm infection did interfere with antigen-specific IFNγ and IL-17 production by splenic T cells, correlating with strongly enhanced production of IL-4 by splenocytes (but not draining lymph node cells). The inhibitory effect of infection on IFNγ appears to be general, antigen- and adjuvant-non-specific, because it was also observed after polyclonal T cell stimulation and independent of the pattern recognition receptor pathway triggered by the adjuvant used. In contrast, the impairment of Th17 induction in the spleen by *N. brasiliensis* was specific for the MINCLE-dependent CAF01. Taken together, immunization experiments in two helminth infection models demonstrated organ-specific inhibition of Th responses in the spleen, with antigen-specific inhibition of CAF01-induced, MINCLE-dependent IL-17 production.

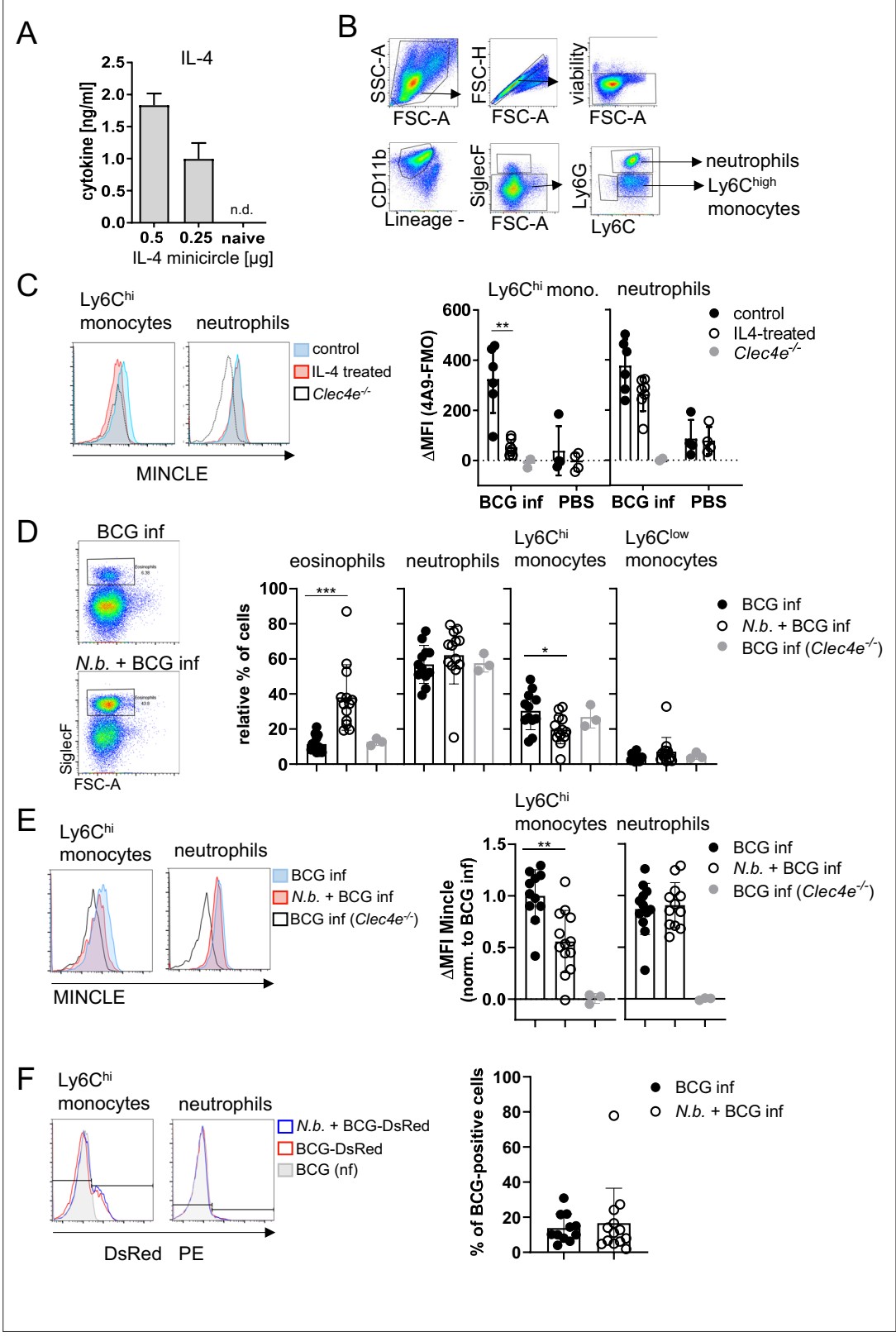

**Figure 3.** Overexpression of IL-4 or co-infection with *Nippostrongylus brasiliensis* impair MINCLE upregulation on peritoneal monocytes but does not reduce phagocytosis upon bacille Calmette-Guerin (BCG) infection. (**A**) IL-4 concentration in serum of mice injected with IL-4 minicircle. C57BL/6 mice were hydrodynamically injected with 0.25 or 0.5 µg of IL-4 plasmid (i.v.) or Ringer solution. Five days (0.5 µg) or 7 days later (0.25 µg) mice were sacrificed

*Figure 3 continued on next page*

*Figure 3 continued*

and serum IL-4 levels were determined via ELISA (n=5–6 mice per group). n.d.=not detectable. (**B, C**) 0.25 µg of IL-4 plasmid or Ringer solution was hydrodynamically injected into C57BL/6 wildtype mice. Two days later mice were infected i.p. with $40 \times 10^6$ CFU of *Mycobacterium bovis* BCG. (**B**) Generic gating strategy for flow cytometry data. Monocytes were characterized as lineage⁻CD11b⁺SiglecF⁻Ly6C⁺ cells. Neutrophils were characterized as lineage⁻CD11b⁺SiglecF⁻Ly6G⁺ cells. Lineage marker: CD3, CD19, NK1.1. (**C**) Histograms depict MINCLE surface expression on Ly6Cʰⁱ monocytes and neutrophils 24 hr p.i. analysed via flow cytometry. Quantitative analysis of MINCLE surface expression shown as median fluorescence intensity (MFI). Fluorescence minus one control (FMO) was substracted. Infected *MINCLE-/-* mice were used as staining controls to exclude unspecific binding of 4A9 antibody. Data is depicted from two independent experiments (2–7 mice per group in total, each dot corresponding to one mouse). **p0.01; ns = not significant. (**D–E**) C57BL/6 mice were s.c. infected with *N. brasiliensis* or left uninfected followed by *M. bovis* BCG-Dsred infection ($40 \times 10^6$ CFU/mouse) on day 10 p.i. Twenty-four hr later mice were sacrificed. (**D**) Representative histograms of eosinophil population of BCG-infected and *N. brasiliensis* co-infected mouse. Bar graphs show relative percentage of myeloid cell populations as indicated. (**E**) MINCLE surface expression was analysed on peritoneal Ly6Cʰⁱ monocytes and neutrophils via flow cytometry. Quantitative analysis of MINCLE surface expression shown as MFI normalized to BCG-infected mice. FMO was substracted. Data is depicted from three independent experiments (11–13 mice per group in total). *p<0.05, **p<0.01, ***p<0.001. (**F**) C57BL/6 mice were s.c. infected with *N. brasiliensis* followed by *M. bovis* BCG-Dsred infection ($40 \times 10^6$ CFU/mouse) or non-fluorescent BCG (nf) on day 10 p.i. Twenty-four hr after BCG infection, phagocytosis was measured by detection of PE signal in monocytes or neutrophils. Quantitative analysis of the percentage of BCG-positive monocytes.

The online version of this article includes the following source data for figure 3:

**Source data 1.** Source data *Figure 3* (cytokine and flow cytometry data).

Finally, we employed mice deficient in IL-4 and IL-13 (designated here as 4-13ko) to determine whether the thwarted MINCLE-dependent Th1/Th17 cytokine responses in helminth-infected mice are caused by the action of these Th2 cytokines. Indeed, in immunization experiments in mice infected with *N. brasiliensis*, we found that the reduced induction of antigen-specific and anti-CD3-triggered production of IFNγ was partially recovered in 4-13ko (*Figure 6A*). The production of IL-17 by spleno-cytes in response to re-stimulation with H1 was not only significantly higher in *N. brasiliensis*-infected 4-13ko than in infected C57BL/6, but in fact exceeded the levels found in non-infected C57BL/6 mice (*Figure 6B*). In contrast, IL-10 production was lower in infected 4-13ko compared to C57BL/6 mice (*Figure 6C*) and no IL-4 was found as expected in the supernatants of 4-13ko splenocytes (*Figure 6D*). Thus, the increased production of IL-4 and/or IL-13 in the spleens of mice infected with *N. brasiliensis* is responsible for the low levels of antigen-specific Th1/Th17 induction by the MINCLE-dependent adjuvant CAF01.

## Discussion

In this study, we show for the first time that IL-4 impairs MINCLE expression and function in vivo. Two different models of helminth infection revealed an organ-specific impairment of Th17 induction by the MINCLE-dependent adjuvant CAF01 in the spleen. These findings establish an in vivo impact of IL-4 on the expression and function of MINCLE that may have important consequences for the detection of and the response to mycobacteria and their cell wall glycolipids in infection or vaccination.

In previous work, we have found that IL-4 downregulates expression of DECTIN-2 family CLRs in murine and human macrophages and DC (*Hupfer et al., 2016*). Here, we first extended these findings by showing that upregulation of these CLR by BCG was also impaired by IL-4 in BMM. Importantly, using overexpression of IL-4 in vivo by hydrodynamic injection of minicircle DNA, the upregulation of MINCLE on monocytes recruited after BCG infection to the peritoneum was strongly inhibited. This effect was cell type-specific, as it was not observed on recruited neutrophils, and it was also caused by infection with *N. brasiliensis*. We have previously shown that the downregulation of MINCLE in mouse macrophages by IL-4 requires the transcription factor STAT6 (*Hupfer et al., 2016*). It is at present unknown which other signalling pathways downstream or in parallel to STAT6 are involved in this process and how they may differ in monocytes/macrophages vs. neutrophils.

IL-4 and *N. brasiliensis* infection did not interfere with phagocytosis of BCG in vitro or in vivo, showing that high-level expression of DECTIN-2 family CLR is not required for phagocytosis of

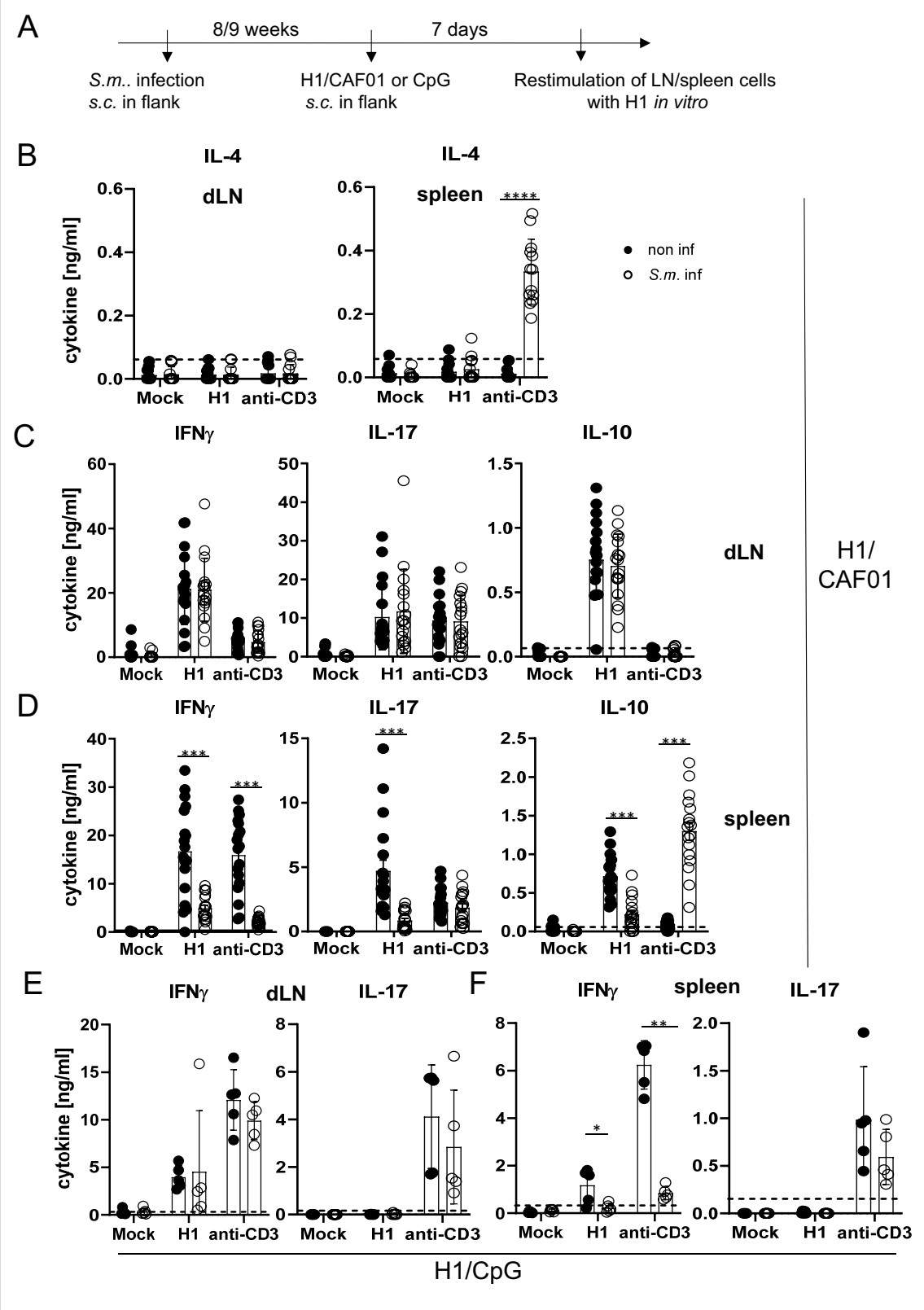

**Figure 4.** Co-infection with *Schistosoma mansoni (S.m.)* suppresses Th1/Th17 induction by a MINCLE-dependent adjuvant in the spleen but not in the draining lymph node. (**A**) Scheme of experimental procedure. *Cercariae* of *S.m.* were injected s.c. into C57BL/6 mice. Eight to 9 weeks p.i. mice were immunized with CAF01 (**B, C, D**) or CpG ODN (**E, F**). Seven days after immunization, mice were sacrificed and inguinal lymph node cells and spleen cells were re-stimulated with H1 in vitro. Draining inguinal lymph node cells or splenocytes were re-stimulated with H1 or anti-CD3 for 96 hr. IL-17, IFNγ, IL-10,

*Figure 4 continued on next page*

*Figure 4 continued*

and IL-4 production was measured from cell culture supernatants by ELISA. Data is shown from three independent experiments (n=18 mice per group in total) for IFNγ, IL-17, and IL-10 (**C, D**) and from two independent experiments (n=13 mice) for IL-4 (**B**). (**E, F**) Data is shown from one experiment (n=5 mice per group in total). *p<0.05, **p<0.01, ***p<0.001. Dashed horizontal lines indicate the limit of detection for each cytokine.

The online version of this article includes the following source data for figure 4:

**Source data 1.** Source data *Figure 4* (cytokine results).

mycobacteria. This may seem surprising, as DECTIN-2 and MCL are phagocytic receptors (*Sato et al., 2006*; *Arce et al., 2004*), and mycobacteria strongly express ligands for both receptors (mannans and TDM). A lack of DECTIN-2 family CLR appears to be compensated by the large number of other receptors contributing to phagocytosis of mycobacteria, including the mannose receptor CD206, integrins like CD11b, or the inhibitory CLR DC-SIGN (*Rajaram et al., 2017*; *Geijtenbeek et al., 2003*; *Melo et al., 2000*).

Importantly, IL-4-treated macrophages not only downregulated DECTIN-2 family CLR, but in addition produced significantly less cytokines after stimulation with BCG. A similar reduction in G-CSF and TNF production was observed here in MINCLE- and FcRγ-deficient BMM. These observations suggests that IL-4 produced during helminth infection could interfere with the initial sensing of invading mycobacteria by pulmonary macrophages and thereby mute the generation of an inflammatory chemokine/cytokine as well as an anti-microbial response. If so, impaired detection of inhaled MTB by alveolar and lung macrophages may favour intracellular mycobacterial survival and replication, providing a mechanism underlying the reported increase in tuberculin skin test conversion reported for helminth-infected household contacts of patients with smear-positive tuberculosis (*Verhagen et al., 2012*). Whether IL-4 impairs functional macrophage responses to mycobacteria primarily through downregulation of MINCLE and DECTIN-2 family CLRs or by inhibiting other cytokine-inducing pathways is an open question that needs to be investigated in future work.

In addition to the response of macrophages to BCG, we investigated how type 2 immune bias during helminth infections impacts on the Th cell differentiation after recombinant subunit vaccination with MINCLE-dependent and -independent adjuvants. Induction of Th17 immunity triggered by the MINCLE-dependent adjuvant CAF01 was suppressed in two models of helminth infection in the spleen, but not in the draining popliteal and inguinal lymph nodes. This organ-specific impact of infection with *N. brasiliensis* and *S. mansoni* on splenic Th cell differentiation was associated with a stronger Th2 bias in the spleen, as demonstrated by the robustly increased levels of IL-4 and IL-10 released by splenocytes, but not draining lymph node cells, from worm-infected mice after polyclonal stimulation with anti-CD3 (*Figures 4 and 5*). Compartmentalized overexpression of IL-10 in the spleens but not lymph nodes has been described before in mice with chronic schistosomiasis (*van der Vlugt et al., 2012*). Our data suggest that such a restriction of Th2 bias to the spleen also applies to IL-4-producing T cells in both helminth infections. For infection with *N. brasiliensis*, we tested mice doubly deficient in *Il4* and *Il13* and confirmed a causal role of IL-4 and/or IL-13 in inhibition of both Th1 and Th17 differentiation in vivo after MINCLE-dependent immunization (*Figure 6*). Similar to the effect of IL-4 on macrophage cytokine responses, it remains to be dissected for the Th1/Th17 adjuvant effect whether the inhibitory effect of IL-4/IL-13 signals is partially or primarily caused by downregulation of DECTIN-2 family CLR in vivo.

Helminth infection did not alter the production of IL-17, IFNγ, or IL-10 by antigen-specific CD4+ T cells in the draining lymph nodes, as determined by re-stimulation of a defined number of cells with recombinant H1. However, the cellularity of the lymph nodes was significantly diminished by *N. brasiliensis*, suggesting that the total number of antigen-specific Th1/Th17 cells in the draining lymph nodes is reduced by co-existent worm infection. Decreased peripheral lymph node size and cellularity has been reported in mice infected with the enteric helminth *H. polygyrus* and was associated with a dampened immune response to infection with BCG, affecting primarily B and T lymphocytes (*Feng et al., 2018*). In the *H. polygyrus* model, the gut-draining mesenteric lymph nodes were increased in size, suggesting that atrophy of peripheral lymph nodes was due to a redistribution of lymphocytes (*Feng et al., 2018*). Furthermore, in mice infected with *Helicobacter pylori*, we have recently shown that concomitant schistosome infection lead to a redistribution of bacterial-specific CXCR3+ T cells resulting in reduced bacterial growth control (*Bhattacharjee et al., 2019*). Cell numbers and size of

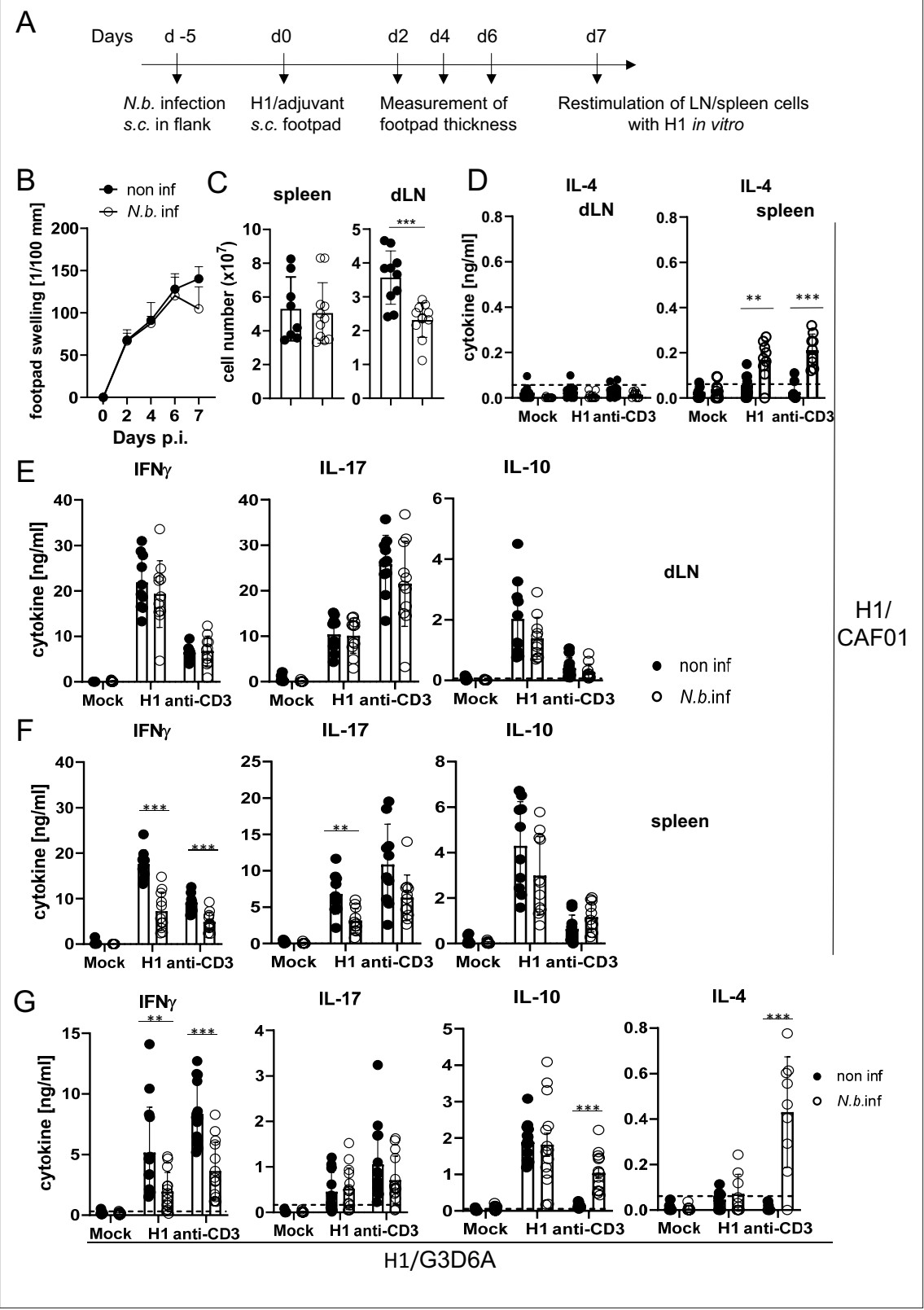

**Figure 5.** Co-infection with *Nippostrongylus brasiliensis* suppresses Th1/Th17 induction by a MINCLE-dependent adjuvant in the spleen but not in the draining lymph node. (**A**) Scheme of experimental procedure. C57BL/6 mice were infected subcutaneously in the flank with 500 L3 *larvae* of *N. brasiliensis* in 200 µl PBS. Five days p.i. mice were immunized with H1/CAF01 (**B–F**) or H1/G3D6A (**G**). Seven days after immunization, mice were sacrificed, and inguinal and popliteal lymph node cells and spleen cells were re-stimulated with H1 in vitro. (**B**) Footpad swelling was measured over

*Figure 5 continued on next page*

*Figure 5 continued*

a period of 1 week after immunization with H1/CAF01 until mice were sacrificed (day 7 after immunization). The increase in footpad swelling is shown as mean + SD for the indicated time points (n=10–11 mice for each data point). (**C**) Absolute cell number of draining inguinal and popliteal lymph node cells (LN) and spleen on day 7 after H1/CAF01. (**D–G**) Draining inguinal and popliteal lymph node cells or splenocytes were re-stimulated with H1, anti-CD3, or left untreated (mock) for 96 hr, followed by cytokine determination by ELISA. (**D**) IL-4 levels produced by draining lymph node cells (left) or splenocytes (right). (**E, F**) IL-17, IFNγ, and IL-10 cytokine production by draining lymph node cells (**E**) and splenocytes (**F**). (**G**) Immunization of C57BL/6 mice with H1 in G3D6A adjuvant. Seven days after immunization mice were sacrificed and splenocytes were treated as described in (**D, E**). All data is shown from two (**B–F**) or three (**G**) independent experiments (n=9–14 mice per group in total). p<0.05, **p<0.01, ***p<0.001. Dashed horizontal lines indicate the limit of detection for each cytokine.

The online version of this article includes the following source data for figure 5:

**Source data 1.** Soruce data *Figure 5* (cytokine results).

lymph nodes are controlled by entry of lymphocytes via the high endothelial venules and exit through the efferent lymphatics.

While the inhibition of IFNγ production by splenocytes from helminth-infected mice was impaired also after immunization with the TLR4 ligand G3D6A and the TLR9 ligand CpG ODN as adjuvants, IL-17 production was specifically impaired when the MINCLE-dependent CAF01 was the adjuvant. In addition, antigen-non-specific production of IFNγ by polyclonal anti-CD3 stimulation was also inhibited in splenocytes from helminth-infected mice, suggesting a general inhibitory effect of *N. brasiliensis* and *S. mansoni* infection that was not observed for production of IL-17. Thus, underlying helminth infection affects the CAF01-induced antigen-specific Th cell response in the spleen in a manner that could be explained by downregulation of MINCLE expression. Whether this is indeed the case needs to be further investigated in future studies comparing myeloid cells in spleen, peripheral lymph node, and subcutaneous injection site tissue. Detection and quantification of MINCLE surface protein on myeloid cells from tissues by flow cytometry is unfortunately complicated by the fact that proteolytic agents commonly used for enzymatic tissue dissociation cause a near-complete loss of specific staining (*Stappers et al., 2021*). For this reason, we employed here the intraperitoneal BCG infection model for assessment of IL-4/helminth effects on myeloid cells in vivo, because it allowed

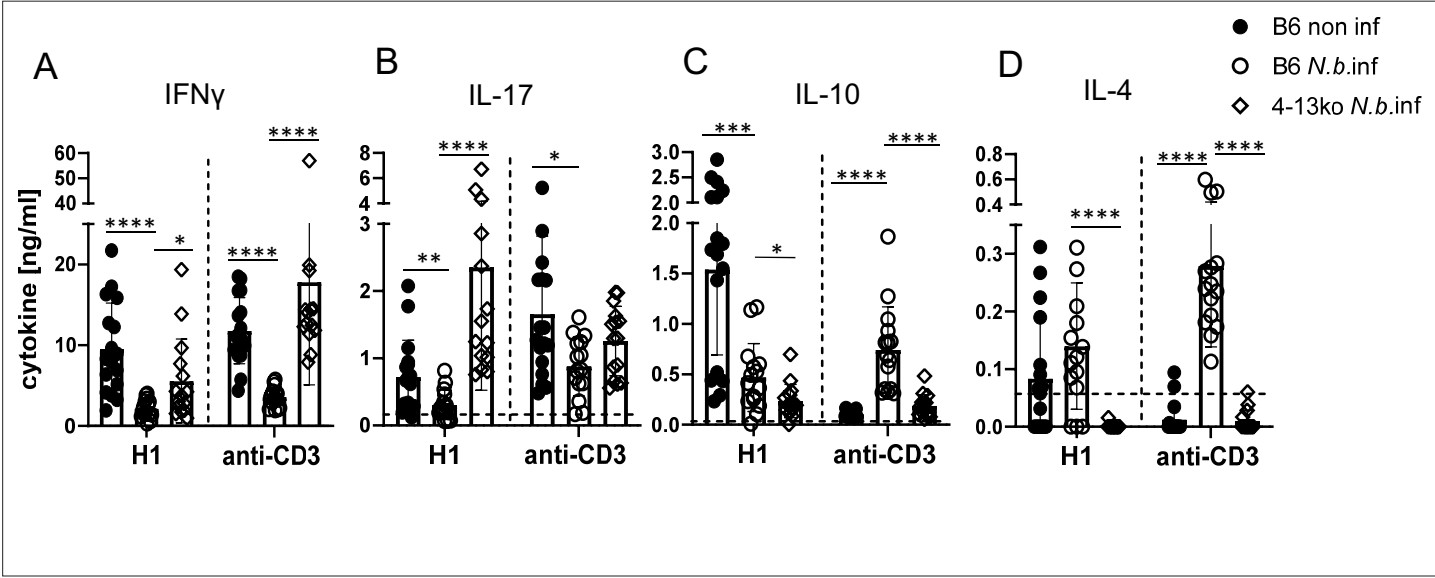

**Figure 6.** Inhibition of Th1/Th17 induction in *Nippostrongylus brasiliensis* infection depends on IL-4 or IL-13. C57BL/6 or of *Il4^-/- Il13^-/-* mice were infected with *N. brasiliensis* or not as indicated and immunized s.c. with H1/CAF01 5 days later. On day 7 after immunization, mice were sacrificed and splenocytes were re-stimulated with H1 or anti-CD3 for 96 hr. Cytokines were measured by ELISA. (**A**) IFNγ, (**B**) IL-17, (**C**) IL-10, (**D**) IL-4. Data shown is pooled from four independent experiments (n=15–17 mice per group in total). p<0.05, **p<0.01, ***p<0.001. Dashed horizontal lines indicate the limit of detection for each cytokine.

The online version of this article includes the following source data for figure 6:

**Source data 1.** Soruce data *Figure 6* (cytokine results).

the direct flow cytometric staining of peritoneal exudate cells without the need for enzymatic tissue digestion.

Interestingly, application of schistosome eggs, a strong inducer of IL-4 and IL-13 production in vivo, prior to infection of mice with *Salmonella typhimurium* downregulated the Th17 response in the gut mucosa and impaired the clearance of the bacteria from the gut (*Schramm et al., 2018*). As the *Salmonella* cell wall contains trehalose phospholipids that bind to and activate MINCLE (*Reinink et al., 2019*), it is possible that induction of IL-17-producing cells during infection requires MINCLE and may be inhibited by schistosome eggs through IL-4/IL-13-induced downregulation of the receptor. Alternatively, IL-4 and IL-10 present in the spleen during helminth infection may act directly on antigen-specific Th cell differentiation after immunization and block Th17 (*Park et al., 2005*; *Harrington et al., 2005*) and Th1 bias (*Paludan, 1998*). Employment of conditional knockout mice with cell type-selective deficiency of the IL-4 receptor in myeloid or T cells in future experiments should be helpful to discriminate between these possibilities in our system of immunization in mice with concurrent helminth infection.

More generally, downregulation of other PRR in addition to DECTIN-2 family CLR during acute or chronic helminth infection may also have a negative impact on vaccines comprising specific PAMPs as adjuvants. Indeed, several papers have demonstrated a diminished baseline expression of TLR2 and TLR9 patients with filarial infection (*Babu et al., 2009a*; *Babu et al., 2005*). Underlying infection with hookworm or filaria were found to downregulate Th1 and Th17 responses in individuals with latent tuberculosis (*Babu et al., 2009b*; *George et al., 2013*). Certainly, helminth-induced immune regulation can interfere by diverse and manifold mechanisms with Th1/Th17 adaptive immunity. It will be worthwhile to investigate further, in animal models and in human patients with helminth infection, whether impaired DECTIN-2 family CLR expression is a significant contributor to thwarting of these protective anti-mycobacterial T cell responses.

## Methods

### Key resources table

| Reagent type (species) or resource | Designation | Source or reference | Identifiers | Additional information |
|---|---|---|---|---|
| Strain, strain background (*Mus musculus*) | C57BL/6 mice | Charles River or Preclinical Experimental Animal Center University Hospital Erlangen | | |
| Strain, strain background (*Mus musculus*) | MINCLE knockout mice | Consortium for Functional Glycomics | *Clec4e*−/− | Wells C et al. 2007J Immunol PMID:18490740 |
| Strain, strain background (*Mus musculus*) | FcRγ knockout mice | Prof. Dr Falk Nimmerjahn, Erlangen, Germany | *Fcer1g*−/− | Takai T et al. 1994Cell PMID:8313472 |
| Strain, strain background (*Mus musculus*) | 4-13ko mice | Prof. AN McKenzie, Cambridge, UK | *Il4*−/− *Il13*−/− | *McKenzie et al., 1999* J Exp Med PMID:10330435 |
| Antibody | Rat monoclonal anti-mouse Mincle (# 4A9) Unlabeled | MBL | MBL-D292-3 | 1:200 dilution |
| Antibody | Rat monoclonal IgG1 Isotype unlabeled | eBioscience | 17-4812-82 | 1:200 dilution |
| Antibody | Rat monoclonal anti-mouse CD3-APC-eF780 | eBioscience | 47-0031-82 | 1:100 dilution |
| Antibody | Rat monoclonal anti-mouse CD19-APC-eF780 | eBioscience | 47-0193-82 | 1:100 dilution |

*Continued on next page*

*Continued*

| Reagent type (species) or resource | Designation | Source or reference | Identifiers | Additional information |
|---|---|---|---|---|
| Antibody | Rat monoclonal anti-mouse NK1.1-APC-eF780 | eBioscience | 47-5941-82 | 1:100 dilution |
| Antibody | Rat monoclonal anti-mouse CD11b-FITC | Biolegend | 101206 | 1:100 dilution |
| Antibody | Rat monoclonal anti-mouse SiglecF-BV421 | BD | 562681 | 1:400 dilution |
| Antibody | Rat monoclonal anti-mouse Ly6C-PerCP-Cy5.5 | Biolegend | 128012 | 1:200 dilution |
| Antibody | Rat monoclonal anti-mouse Ly6G-PE-Cy7 | Biolegend | 127618 | 1:400 dilution |
| Antibody | Human monoclonal anti-mouse Dectin1-APC | Miltenyi | 130-102-250 | 1:50 dilution |
| Antibody | Human monoclonal anti-mouse Dectin2-APC | Miltenyi | 130-116-911 | 1:50 dilution |
| Antibody | Human IgG1 APC (Isotype for Dectin2, Dectin1) | Miltenyi | 130-113-446 | 1:50 dilution |
| Chemical compound, drug | Mouse Fixable viability dye eF506 | eBioscience | 65-0866-18 | 1:1000 dilution |
| Antibody | Rat IgG1 APC | eBioscience | 17-4301-81 | 1:200 dilution |
| Antibody | Rat IgG1 APC Iso k | eBioscience | 14-4301-82 | 1:200 dilution |
| Sequence-based reagent | Mincle Primer (qRT-PCR) | Metabion | | For: gctcacctggtggttatcg<br>Rev: aggttttgtgcgaaaaagga |
| Sequence-based reagent | Mcl Primer (qRT-PCR) | Metabion | | For: agtaacgtgcatccgagagg<br>Rev: taacaggacagcaggtccaa |
| Sequence-based reagent | Dectin2 Primer (qRT-PCR) | Metabion | | For: cagtgaagggactatggtgtca<br>Rev: ggagccaaatgacttccagt |
| Sequence-based reagent | Dectin1 Primer (qRT-PCR) | Metabion | | For: atggttctgggaggatggat<br>Rev: atggttctgggaggatggat |
| Sequence-based reagent | Hprt Primer (qRT-PCR) | Metabion | | For: tcctcctcagaccgctttt<br>Rev: cctggttcatcatcgctaatc |
| Sequence-based reagent | Mincle Probe | Roche | Roche Universal Probe Library (UPL) | #15 |
| Sequence-based reagent | Mcl Probe | Roche | Roche UPL | #1 |
| Sequence-based reagent | Dectin2 Probe | Roche | Roche UPL | #89 |
| Sequence-based reagent | Dectin1 Probe | Roche | Roche UPL | #60 |
| Sequence-based reagent | Hprt Probe | Roche | Roche UPL | #95 |
| Recombinant DNA reagent | Minicircle DNA encoding IL-4 | Dr Stefan Wirtz, Erlangen, Germany | | |
| Strain, strain background (bacteria) | *Mycobacterium bovis* BCG DsRed Danish | Dr Anca Dorhoi, Friedrich Löffler Institute, Greifswald, Germany | | |

*Continued on next page*

*Continued*

| Reagent type (species) or resource | Designation | Source or reference | Identifiers | Additional information |
|---|---|---|---|---|
| Strain, strain background (helminth) | *Nippostrongylus brasiliensis* | Dr David Vöhringer, Erlangen, Germany | | 500 L3 larvae per mouse s.c. |
| Strain, strain background (helminth) | *Schistosoma mansoni* | Dr Clarissa Prazeres da Costa, Munich, Germany | NMRI strain | 100 cercariae per mouse s.c. |
| Other | CpG ODN1826 | TIB MOLBIOL | 180000237 | 0.5 µM (in vitro); 10 nmol per mouse in vivo |
| Other | CAF01 (TDB+DDA liposomes) | Dr Dennis Christensen, Statens Serum Institute, Copenhagen, Denmark | | Adjuvant, 50 µl injected s.c. together with H1 protein |
| Other | G3D6A liposomal formulation | Dr Christian Alexander, Research Center Borstel | | Adjuvant, 50 µl injected s.c. together with H1 protein |
| Peptide, Recombinant protein | H1 Mycobacterial antigen | Dr Dennis Christensen, Statens Serum Institute, Copenhagen, Denmark | | H1 is a fusion protein of Ag85B and ESAT-6. 2 µg H1 in 100 µl CAF01 for immunization and 1 µg/ml for re-stimulation |
| Peptide, Recombinant protein | Recombinant murine IL-4 | PeproTech | 214–14 B | 10 µg/ml for in vitro stimulation |
| Peptide, Recombinant protein | Purified anti-CD3 e | Biolegend | 100302 | 0.5 µg/ml re-stimulation |

## Mice

C57BL/6 wildtype, MINCLE knockout (*Clec4e*[-/-]), FcRγ knockout (*Fcer1g*[-/-]), and *Il4*[-/-] *Il13*[-/-] mice were bred under specific pathogen-free conditions at the 'Präklinische Experimentelle Tierzentrum' (PETZ) of the Medical Faculty in Erlangen. *Clec4e*[-/-] mice were generated by the Consortium for Functional Glycomics (*Wells et al., 2008*) and used with permission. *Fcer1g*[-/-] mice (*Takai et al., 1994*) were kindly provided by Dr Falk Nimmerjahn. *Il4*[-/-] *Il13*[-/-] were originally provided by AN McKenzie (MRC Laboratory of Molecular Biology, Cambridge, UK) and backcrossed to CD45.1_C57BL/6 background for more than 10 generations (*McKenzie et al., 1999*). C57BL/6N mice were purchased from Charles River Laboratories. All mouse experiments were approved by the 'Regierung von Unterfranken' (protocol number 55.2.2-2532-543) and the 'Regierung of Oberbayern' (protocol number ROB-55.2 Vet_02-17-145). Male mice between 8 and 12 weeks of age were used for in vivo experiments.

## Bacteria

*M. bovis* (BCG) was grown in Middlebrook 7H9 broth supplemented with 10% OADC-enrichment medium and 0.05% Tween 80 in small cell culture flasks constantly shaking at 125 rpm at 37°C to an $OD_{600}$ ~1–2. Prior to in vitro stimulation, BCG were washed with PBS and diluted in cDMEM.

## Hydrodynamic injection of IL-4 minicircle DNA

To investigate IL-4-derived effects on immunization responses or BCG infections in mice, we utilized a technique that leads to a systemic overexpression of IL-4. A minicircle DNA vector that encodes the gene for IL-4, but lacks further bacterial elements in comparison to conventional plasmids, was hydrodynamically injected intravenously in the tail vein of 5-week-old male mice (*Liu et al., 1999*); 0.5 µg of IL-4 was injected in 2 ml Ringer solution in under 10 s per mouse.

## Helminth infections

All *S. mansoni* infection experiments were performed at TU Munich. C57BL/6 mice were infected subcutaneously as described previously (*Bhattacharjee et al., 2019*) with 100 cercariae from the NMRI strain (originally from Puerto Rico) of *S. mansoni* in 100 µl PBS shed by infected *Biomphalaria*

*glabrata* snails, provided by the NIAID Schistosomiasis Resource Center of Biomedical Research Institute (Rockville, MD) through NIH-NIAID Contract HHSN272201700014I for distribution through BEI Resources. Mice were vaccinated after 8 weeks of infection, when parasite egg-induced Th2 immune responses begin to peak.

The *N. brasiliensis* life cycle was maintained in rats. *N. brasiliensis* larvae were cultured in a mixture of charcoal and feces of infected rats in petri dishes for 15–40 days at room temperature. Subsequently, third-stage (L3) larvae were harvested with 10 ml of 0.9% saline and transferred to a Baermann apparatus. Within 1 hr larvae descended to the bottom of the funnel. Larvae were then transferred to a fresh 50 ml tube and extensively washed by repeating steps of sedimentation and exchange of saline. C57BL/6 mice were infected subcutaneously with 500 L3 larvae in 200 µl PBS in the flank of each mouse using a 25 G needle. All worms were expelled completely within 10 days.

### BCG infection of mice

Ten days after *N. brasiliensis* infection, or 2 days after hydrodynamic injection of IL-4 minicircle DNA, mice were infected intraperitoneally with $40 \times 10^6$ CFU of *M. bovis* BCG in a volume of 200 µl PBS using a 27 G needle. PBS-injected mice as well as completely naïve mice were used as controls. Mice were sacrificed 24 hr post infection.

### Immunizations

Mice were immunized subcutaneously with 50 µl of a mixture of 1 µg H1, a fusion protein of the MTB antigens Ag85B and ESAT-6, and the respective adjuvants (CAF01, CpG ODN 1826, or G3D6A) in the footpads of the hind legs or in the flank as indicated in the figure legends. CAF01 is composed of TDB and cationic dimethyldioctadecylammonium (DDA) liposomes and has been described in detail before (*Agger et al., 2008*). CpG ODN 1826 is a phosphorothioate-protected oligonucleotide and was synthesized by TIB MOLBIOL (Berlin, Germany). G3D6A is a liposomal adjuvant formulation. It is comprised of the synthetic TLR4 ligand 3-*O*-de-acyl-hexaacyl-monophoshoryl lipid A (3D-6A-SMPLA, 3D(6-acyl)-PHAD) manufactured as cGMP product by Avanti Polar Lipids Inc (Alabaster, AL, USA) and embedded in a matrix of 1,2-dimyristoyl-*sn*-glycero-3-phosphocholine, 1,2-dimyristoyl-*sn*-glycero-3-phosphoglycerol and cholesterol (molar ration: 9:1:7.5) in an aqueous suspension buffered with PBS. Footpad swelling was monitored regularly and measured prior to immunization as well as every second day post immunization. On day 7 post immunization, inguinal and popliteal lymph nodes were analysed.

### Isolation and culture of BMM

Bone marrow cells were differentiated to BMM for 6–7 days in complete Dulbecco's modified Eagle medium containing 10% FCS, 50 µM β-mercaptoethanol, and penicillin/streptomycin (cDMEM) supplemented with 10% L929 cell-conditioned medium as a source of M-CSF.

### Stimulation of BMM

BMM were stimulated with plate-coated TDB (Polar Avanti, 5 µg/ml), using isopropanol as a mock control, as described (*Werninghaus et al., 2009*), LPS (*Escherichia coli* serotype O55:B5, Sigma, 10 ng/ml), CpG ODN 1826 (TIB MOLBIOL, 0.5 µM), or with BCG at the indicated MOI.

### Re-stimulation of lymph node cells

Inguinal and popliteal lymph nodes were collected and meshed through a 70 µm nylon filter to get a single-cell suspension. $5 \times 10^5$ cells were stimulated in 96-well U-bottom plates with H1 (1 µg/ml), soluble anti-CD3 (0.5 µg/ml), or left untreated (mock) for 96 hr.

### mRNA expression of CLR

RNA was isolated using Trifast (Peqlab) and transcribed to cDNA (High capacity cDNA Reverse Transcription Kit, Applied Biosystems). Expression of the house keeping gene *Hprt* and of the genes of interest was analysed by qRT-PCR. All primers and probes were selected from the Roche Universal Probe Library. Ct values of the target genes were normalized to *Hprt*, calibrated to unstimulated cells, and depicted as fold change.

## Cytokine ELISA

Secreted cytokines were analysed by ELISA (R&D Systems) from cell culture supernatants of stimulated BMM or lymph node cells.

## Flow cytometry

Cell surface expression of MINCLE was analysed by flow cytometry. Cells were blocked with anti-mouse CD16/32, stained with the primary antibody anti-MINCLE (clone 4A9, MBL, 1 µg/ml), followed by anti-rat IgG1-APC (eBioscience). FACS data were acquired on an LSRFortessa (BD) and analysed using the software FlowJo (v10).

## Statistics

GraphPad Prism software (version 8) was used for statistical analysis. Statistical significance was calculated using Mann-Whitney U-test to compare two non-paired groups or Wilcoxon signed rank test for paired groups. *$p<0.05$, **$p<0.01$, ***$p<0.001$, ns $p>0.05$.

## Acknowledgements

Animal husbandry by Manfred Kirsch, technical assistance by Nina Grohmann, and support by Christian Bogdan are gratefully acknowledged. We thank Nathalie Thuma and Paul Haase for help with *N. brasiliensis* infections. This work was funded by Deutsche Forschungsgemeinschaft (GRK 1660-TP-A02 and LA 1262/8-1 to RL, and CRC1181_A02 to DV, and CO 1469/16-1 to CPdC).

## Additional information

### Funding

| Funder | Grant reference number | Author |
|---|---|---|
| Deutsche Forschungsgemeinschaft | LA 1262/8-1 | Roland Lang |
| Deutsche Forschungsgemeinschaft | CDC 1181_A02 | David Voehringer |
| Deutsche Forschungsgemeinschaft | CO 1469/16-1 | Clarissa Prazeres da Costa |

The funders had no role in study design, data collection and interpretation, or the decision to submit the work for publication.

### Author contributions

Judith Schick, Formal analysis, Validation, Investigation, Visualization, Writing – review and editing; Meltem Altunay, Formal analysis, Investigation, Visualization, Writing – review and editing; Matthew Lacorcia, Formal analysis, Investigation, Writing – review and editing; Nathalie Marschner, Julia Schluckebier, Barbara Bodendorfer, Investigation; Stefanie Westermann, Christian Alexander, Stefan Wirtz, Resources; Christoph Schubart, Resources, Methodology; Dennis Christensen, Resources, Writing – review and editing; David Voehringer, Formal analysis, Supervision, Funding acquisition, Writing – review and editing; Clarissa Prazeres da Costa, Resources, Formal analysis, Supervision, Writing – review and editing; Roland Lang, Conceptualization, Formal analysis, Supervision, Funding acquisition, Visualization, Writing - original draft, Project administration, Writing – review and editing

### Author ORCIDs

Stefan Wirtz ⓘ http://orcid.org/0000-0001-6936-7431
David Voehringer ⓘ http://orcid.org/0000-0001-6650-0639
Roland Lang ⓘ http://orcid.org/0000-0003-0502-3677

### Ethics

All mouse experiments were approved by the "Regierung von Unterfranken" (protocol number 55.2.2-2532-543) and the "Regierung of Oberbayern" (protocol number ROB-55.2 Vet_02-17-145).

## Decision letter and Author response
Decision letter https://doi.org/10.7554/eLife.72923.sa1
Author response https://doi.org/10.7554/eLife.72923.sa2

## Additional files

### Supplementary files
• Transparent reporting form

### Data availability
For most figures, the data are directly shown as individual data points. These quantitative data are provided in the accompanying Excel files "Source data Schick et al.xls".

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
