## [Editor Report]

The effect of helminth infection on vaccination against tuberculosis infection and disease is an important area of study. In this manuscript, the authors build off of a large body of prior data showing that mycobacterial antigens upregulate MINCLE whilst the cytokine IL-4 downregulates MINCLE, and as IL-4 is upregulated during Helminth infections, this can antagonize Th1/Th17 responses. By using two different models of helminth infection, the authors demonstrate compelling organ-specific impairment of Th17 responses in a vaccination setting with a MINCLE-dependent adjuvant. The work is topical, may have important translational implications for patients with tuberculosis and helminth co-infections and/or vaccination regimens for patients with helminth infections, and will be of interest to individuals studying the convergence of different infectious diseases.

---

## [Decision Letter]

**Decision letter after peer review:**

Thank you for submitting your article "IL-4 and helminth infection downregulate MINCLE-dependent macrophage response to mycobacteria and Th17 adjuvanticity" for consideration by *eLife*. Your article has been reviewed by 3 peer reviewers, one of whom is a member of our Board of Reviewing Editors, and the evaluation has been overseen by Bavesh Kana as the Senior Editor. The reviewers have opted to remain anonymous.

Essential revisions:

1) The authors conclude that the organ-specific effects on the Th17 vaccination response in the helminth infection setting are mediated by IL-4 (see, for instance, abstract, l. 48), tying together the different parts of the manuscript. The authors should provide further evidence for a direct contribution of IL-4 to the thwarted Th17 response to MINCLE-dependent immunization in these helminth models. For instance, interference with IL-4 secretion (e.g. by IL-4 blockade) would markedly strengthen the impact of the manuscript.

2) The kinetics of the IL-4-mediated effects on MINCLE expression and function would be helpful to understand the biological implications. For instance, in the in vitro studies, have the authors performed IL-4 treatment at different time points (e.g. pre-incubation)?

3) In Figure 1, the authors observed that a similar pattern of induced expression and regulation by IL-4 as for MINCLE was observed for the CLRs DECTIN-2 and MCL as well. Have the authors elaborated on this finding more in depth? Why did the authors focus exclusively on MINCLE?

4) It would be interesting for the authors to add some discussion as to why the IL-4-induced downregulation of Mincle was not observed in neutrophils.

5) For Figure 3E, did the authors examine the cytokine production from lymphnode cells or splenocytes upon re-stimulation with mycobacterial antigens?

6) Line 173- all data should be shown, especially early timepoints in this case since 24 hours is too late to measure phagocytosis. An even earlier time point than 6 hours would also be beneficial.

7) Figure 3 and Lines 203-204- The differences in eosinophils is very obvious, but since all cell types are shown as frequencies, the reduced frequency of monocytes/macrophages could be due to the increased eosinophils. In order to make any conclusions about the number of macrophages/monocytes, total cell numbers must be provided and compared.

8) Line 177- It is possible that the MINCLE deficient macrophages have a defect in phagocytosis that leads to lower levels of TNF and G-CSF. The authors should present phagocytosis and bacterial burden data for the Mincle-/- macrophages to be able to discuss a potential role (or lack thereof) for bacterial number.

9) In the material and methods, the authors mention the BCG MOI as "indicated in figure " but did not provide the MOI in figure 1 A-D.

10) Figure 3 claims to address: "Co-infection with N.brasiliensis (hookworm) impairs MINCLE upregulations on peritoneal monocytes but does not reduce phagocytosis during BCG infection", but Figure 3 does not test phagocytosis.

11) Figure 3D and E would benefit with an N.b. control to see how just the helminth infection affects cell populations.

12) The graphs in figure 4 (i.e. 4B) look like they should be on a log scale. On the linear scale, the higher points completely wash out any difference in the lower points. Also be sure to mention the limit of detection, were these all above it? This comment also applies to similar panels in Figure 5.

13) The authors repeatedly claim to be measuring "T cell response", but they never isolate T cells and the experiments were performed with total splenocyte or total lymph node cells.

14) In the figures, the authors should specify H1/CAF and H1/CpG, instead of both being a common H1 designation. i.e. 4A-D: H1/CAF and E,F: H1/CpG.

---

## [Author Response]

Essential revisions:1) The authors conclude that the organ-specific effects on the Th17 vaccination response in the helminth infection setting are mediated by IL-4 (see, for instance, abstract, l. 48), tying together the different parts of the manuscript. The authors should provide further evidence for a direct contribution of IL-4 to the thwarted Th17 response to MINCLE-dependent immunization in these helminth models. For instance, interference with IL-4 secretion (e.g. by IL-4 blockade) would markedly strengthen the impact of the manuscript.

We thank the Reviewers for this suggestion to test a causal role of IL-4 in the inhibition of the Th17 response.

As an alternative strategy to investigate whether IL-4 directly contributes to inhibition of the Th17 immune response in vivo we employed mice deficient in both Il4 and Il13 genes (designated 4-13ko mice). We chose this genetic tool because we have previously found that both IL-4 and IL-13 down-regulate MINCLE expression on macrophages in vitro through STAT6 signaling (Hupfer et al. 2016 Front Immunol) and both cytokines are expressed during helminth infection.

Following immunization with the MINCLE-dependent adjuvant CAF01, the reduced production of IFNγ and IL-17 found in splenocytes of C57BL/6 mice with concurrent infection with N. brasiliensis was restored in the 4-13ko mice. Thus, IL-4 and/or IL-13 are causally linked to the loss of Th1/Th17 adjuvanticity in mice with underlying helminth infection.

These new data are shown in new Figure 6 and in the Results section (lines 307-319) in the revised manuscript, including the results for IL-10 (lower in infected 4-13ko mice) and IL-4 (as expected not produced in 4-13ko mice). These findings are also described in the Abstract (lines 49-52) and are interpreted in the Discussion section (lines 370-375; lines 414-417).

2) The kinetics of the IL-4-mediated effects on MINCLE expression and function would be helpful to understand the biological implications. For instance, in the in vitro studies, have the authors performed IL-4 treatment at different time points (e.g. pre-incubation)?

The Reviewers’ comment raises an interesting question because in vivo the timing of stimulation of APC with the MINCLE-triggering adjuvant may be after, concomitantly or before being exposed to IL-4. However, in all in vitro experiments performed for this manuscript, we did add the IL-4 at the time of stimulating the macrophages with BCG or LPS. On the other hand, the down-regulation of MINCLE on monocytes in vivo by IL-4 via minicircle DNA injection constitutes a pre-exposure setting. Therefore, it appears that concomitant exposre to and pre-incubation with IL-4 have comparable effects.

3) In Figure 1, the authors observed that a similar pattern of induced expression and regulation by IL-4 as for MINCLE was observed for the CLRs DECTIN-2 and MCL as well. Have the authors elaborated on this finding more in depth? Why did the authors focus exclusively on MINCLE?

There are two reasons for focusing on MINCLE in this manuscript: first, it showed the strongest inducibility by stimulation with TDB or BCG, and, second, the in vivo immunization experiments were done using the MINCLE-dependent adjuvant CAF01 (Figures 4 and 5). However, we agree with the Reviewers that the regulation of DECTIN-2 and MCL, and also of DECTIN-1, by IL-4 are also interesting.

Therefore, we have performed flow cytometry to validate the mRNA expression data shown in Figure 1B and D for DECTIN-2 and for DECTIN-1. Unfortunately, MCL protein levels could not be investigated because we could not get our hands on a suitable antibody.

The new data we obtained confirm at the cell surface protein level the upregulation of DECTIN-2 after stimulation with BCG and its inhibition by IL-4. In addition, these data also show that IL-4 alone induces DECTIN-1 upregulation at the protein level; however, stimulation with BCG led to downregulation of DECTIN-1 at the cell surface, contrary to what was observed at the mRNA level (Figure 1D). These new data are included as new Figure 1F and 1G and described in the Results section (lines 171-175) of the revised manuscript.

4) It would be interesting for the authors to add some discussion as to why the IL-4-induced downregulation of Mincle was not observed in neutrophils.

This is an interesting and difficult question. Perhaps neutrophils are less transcriptionally responsive compared to monocytes/macrophages because of their short life span. Definitely, it would be very interesting to compare side-by-side the response to IL-4 between neutrophils and macrophages in a future experiment. We have commented on this point in the Discussion (lines 335-337).

5) For Figure 3E, did the authors examine the cytokine production from lymphnode cells or splenocytes upon re-stimulation with mycobacterial antigens?

Figure 3E shows the expression of MINCLE on peritoneal monocytes and neutrophils. Therefore, we assume that the question is in fact directed at Figure 4E/F, where cell suspensions from lymph nodes or spleens were re-stimulated with anti-CD3 antibody or the fusion protein H1 in vitro. In fact, H1 is a fusion of two proteins from *M. tuberculosis* (Ag85B and ESAT-6), thus, we did exactly what the Reviewers are asking for.

6) Line 173- all data should be shown, especially early timepoints in this case since 24 hours is too late to measure phagocytosis. An even earlier time point than 6 hours would also be beneficial.

As requested by the Reviewers, we include now the phagocytosis data for the 6 hour timepoint, that were mentioned before as data not shown, in Figure 2A. The histograms and quantitation look very similar to what was already depicted for the 24 hour timepoint. We have not performed phagocytosis assays at earlier timepoints for IL-4-treated macrophages because any changes that may occur in phagocytosis capacity in the period before 6 hours will be transient given the absence of an IL-4 effect after 6 and 24 hours, and we do not think that such transient effects would be functionally relevant in terms of the impact of IL-4 on the cytokine response to BCG that is analyzed in Figure 2, too.

Comment (8) below addresses a related topic, namely the question whether phagocytosis is dependent on Mincle or the adapter protein FcRγ. This question is relevant because we have used BMM deficient in these proteins as comparison to IL-4-treatment and the impact on cytokine production in Figure 2. Indeed, we have determined phagocytosis in assays using BMM from Mincle^-/-^ and Fcer1g^-/-^ mice that showed no alterations in the uptake of BCG. These results are now included as new Figure 2C.

7) Figure 3 and Lines 203-204- The differences in eosinophils is very obvious, but since all cell types are shown as frequencies, the reduced frequency of monocytes/macrophages could be due to the increased eosinophils. In order to make any conclusions about the number of macrophages/monocytes, total cell numbers must be provided and compared.

We agree that no statement can be made here about the numbers of macrophages or monocytes. In fact, our intention here was primarily to examine the impact of helminth infection on the expression of MINCLE on these cells. Therefore, we did not regularly count the absolute numbers of peritoneal cells. However, in the two experiments when cells were counted, there was no difference in total peritoneal exudate cell numbers between the conditions “BCG alone” and “BCG+N. brasiliensis”.

**Author response image 1. sa2fig1:** 

8) Line 177- It is possible that the MINCLE deficient macrophages have a defect in phagocytosis that leads to lower levels of TNF and G-CSF. The authors should present phagocytosis and bacterial burden data for the Mincle-/- macrophages to be able to discuss a potential role (or lack thereof) for bacterial number.

The question whether phagocytosis is dependent on Mincle or the adapter protein FcRγ is relevant because we have used BMM deficient in these proteins as comparison to IL-4-treatment and the impact on cytokine production in Figure 2. Indeed, we have determined phagocytosis in assays using BMM from Mincle^-/-^ and Fcer1g^-/-^ mice that showed no alterations in the uptake of BCG. These results are now included as new Figure 2C in the revised manuscript.

9) In the material and methods, the authors mention the BCG MOI as "indicated in figure " but did not provide the MOI in figure 1 A-D.

This information (MOI 10) is now given in the legend to Figure 1.

10) Figure 3 claims to address: "Co-infection with N.brasiliensis (hookworm) impairs MINCLE upregulations on peritoneal monocytes but does not reduce phagocytosis during BCG infection", but Figure 3 does not test phagocytosis.

The impact of infection with N. brasiliensis on phagocytosis of BCG in vivo was determined by flow cytometry of peritoneal lavage cells harvested 24 hours after infection with BCG-DsRed. As shown in Figure 3F, the percentages of BCG-containing monocytes were comparable in mice with underlying helminth infection and controls.

11) Figure 3D and E would benefit with an N.b. control to see how just the helminth infection affects cell populations.

N. brasiliensis infection without injection of BCG was only included in one of the experiments with two mice. Given this low number of replicates, we decided not to include the data in Figure 3 D+E. As expected the data from this single experiment showed a dominance of eosinophils in the peritoneum of mice infected with N. brasiliensis.

12) The graphs in figure 4 (i.e. 4B) look like they should be on a log scale. On the linear scale, the higher points completely wash out any difference in the lower points. Also be sure to mention the limit of detection, were these all above it? This comment also applies to similar panels in Figure 5.

We also considered the question how best to scale the y-axis. Finally, we decided to keep the linear scale. The low values were close to the limit of detection; these are now depicted in the graphs of Figures 4, 5 and 6, or stated in the Figure legends. The distribution of the data is quite wide; however, the statistical testing and average values demonstrate that significant differences can be observed in helminth-infected mice compared to controls.

13) The authors repeatedly claim to be measuring "T cell response", but they never isolate T cells and the experiments were performed with total splenocyte or total lymph node cells.

The Reviewers correctly point out we do not show in vitro restimulations with isolated T cells or perform intracellular cytokine stainings to demonstrate T cells are the source of the cytokines measured in the supernatants of lymph node and spleen cells after re-stimulation with H1 or anti-CD3 (Figure 4 and Figure 5). We have addressed this question in earlier work published in 2013, showing that IFNγ and IL-17 are detectable by ICS mostly in CD4^+^ T cells (Desel C et al. PLoS ONE 8: e53531, doi:10.1371/journal.pone.0053531, Supplemental Figure S1). This paper is cited in the manuscript as reference 51 (line 238). The group of Peter Andersen has published several papers comprehensively analyzing the antigen-specific T cell response using exactly the same antigen-adjuvant combination as in this manuscript. Their data demonstrate the development of multifunctional antigen-specific CD4^+^ T cells producing IFNγ, IL-2, and TNF (Lindenstrom T et al. J Immunol 182: 8047. doi.org/10.4049/jimmunol.0801592); in a recent publication combining the CAF01 formulation with a Chlamydia vaccine, the antigen-specific production of IL-17 by CD4^+^ T cells was demonstrated by ICS (Nguyen N et al. NPJ Vaccines 5: 7. doi: 10.1038/s41541-020-0157-x. eCollection 2020). Thus, our notion that the cytokines secreted during the re-stimulation assays in vitro are T cell-derived is based on published data from several groups for IFNγ and IL-17. These papers are now cited in the Results section (lines 235-239).

14) In the figures, the authors should specify H1/CAF and H1/CpG, instead of both being a common H1 designation. i.e. 4A-D: H1/CAF and E,F: H1/CpG.

The use of the different adjuvants CAF01 or CpG or G3D6A is now indicated on the respective panels in Figures 4 and 5, as suggested by the Reviewers.